# Impact of Microorganisms and Parasites on Neuronally Controlled *Drosophila* Behaviours

**DOI:** 10.3390/cells10092350

**Published:** 2021-09-08

**Authors:** Martina Montanari, Julien Royet

**Affiliations:** Aix-Marseille Université, CNRS, IBDM, 13288 Marseille, France; martina.montanari@univ-amu.fr

**Keywords:** host–pathogen interactions, behavioral immunity, *Drosophila melanogaster*, neurons, bacteria, parasitoid wasps, NF-κB, octopamine, antimicrobial peptides

## Abstract

Like all invertebrates, flies such as *Drosophila* lack an adaptive immune system and depend on their innate immune system to protect them against pathogenic microorganisms and parasites. In recent years, it appears that the nervous systems of eucaryotes not only control animal behavior but also cooperate and synergize very strongly with the animals’ immune systems to detect and fight potential pathogenic threats, and allow them to adapt their behavior to the presence of microorganisms and parasites that coexist with them. This review puts into perspective the latest progress made using the *Drosophila* model system, in this field of research, which remains in its infancy.

## 1. Introduction

Insects live in diversified ecological niches that, while extremely variable in terms of physical and chemical characteristics, are all colonized by microorganisms such as bacteria, viruses, fungi, and also by parasites. It follows that, from the earliest stages of their development until their death, animals interact for better or for worse with these co-inhabitants [1]. For the better, as it is now well demonstrated, microbes can positively impact various physiological parameters of the host such as fecundity, longevity, and growth, to name but a few [2,3,4,5]. For worse, since, obviously, some of these microbes and parasites can negatively affect the host and can even sometimes be life-threatening for them. To defend themselves, insects have developed immune strategies to identify surrounding microorganisms and trigger ad hoc cellular and humoral responses that eradicate invaders and ensure the physical integrity and fitness of the host and its progeny [6,7,8,9]. Preceding studies highlight the benefits of bringing neurons into the complex host–microbe interaction game [10]. Sensory neurons play a role in identifying microbes and, thus, in distinguishing beneficial ones to live with from other, potentially pathogenic, ones to avoid. In contrast, host neurons can be hijacked by microorganisms and microbe-derived products to ease their proliferation within infected animals. In addition, the nervous system’s perception of a microbial threat may allow the host to modify its behavior to reduce the consequences of the infestation on itself and its offspring. Some of these mechanisms have been described under the generic term of behavioral immunity [11]. As neuroscientists and immunologists continue to uncover molecules acting across both systems and genetic interactions between them, it becomes clear that the immune and the neuronal systems share many components, and cooperate at many different levels to allow an animal to live in harmony with its exogenous and endogenous microbes and parasites [12]. *Drosophila melanogaster,* with its powerful collection of genetic and genomic tools, has been an outstanding platform to identify components and discover new mechanisms and paradigms regulating both the neuronal and the immune systems [13]. More recently, the fruit fly has been used to study how immune and neuronal mechanisms cooperate to enable flies to protect themselves from pathogenic microbes, and sometimes to take advantage of the microorganisms and parasites they live with. They may even co-evolve. Here, we illustrate how fly studies are starting to help assemble some pieces of this highly intricate puzzle, of which many pieces are still missing.

## 2. A Brief Resumé of the *Drosophila* Immune System and its Players

Like other invertebrates, *Drosophila* species do not possess an adaptive immune system and only rely on innate mechanisms to fight pathogens. This antimicrobial response is mediated by both cellular and humoral mechanisms that include activation of signal transduction pathways, phagocytosis, wound healing, and the encapsulation of previously melanized foreign elements [14]. The humoral immune response, activated upon microorganism’s detection, triggers the production and release of a cocktail of immune regulators and effectors such as antimicrobial peptides (AMPs) into the circulating hemolymph. This response is mainly driven by two evolutionarily conserved NF-κB-dependent pathways called immune deficiency (IMD) and Toll, whose upstream pattern recognition receptors are members of the peptidoglycan recognition protein family (PGRP) [15,16]. The Toll pathway is induced after the detection of bacteria Lys-type peptidoglycan (PGN) by the circulating PGRP-SA sensor [17]. The IMD pathway is induced by bacteria containing DAP-type peptidoglycan that is detected either by the cytosolic PGRP-LE or the membrane tethered PGRP-LC [18,19,20] (Figure 1). Gram-positive bacteria possess Lysine-type PGN whereas Diaminopimelic-type PGN is found mainly in the cell wall of Gram-negative bacteria and bacilli. These pathways culminate in the translocation of NF-κB homo- and heterodimers to the nucleus leading to infection-specific upregulation of AMPs targeted at clearing the infection. The response to virus replication within the host cells involves both the JAK-STAT (Janus kinase/signal transducer and activator of transcription) pathway and the RNA interference machinery, and somewhat overlaps with the genes induced upon bacterial and fungal infections, such as NF-κB signaling [21]. The cell-mediated immune response is induced upon epithelial damage and following the detection of foreign particles, and it involves three cell types [22]. The plasmatocytes represent the majority of the circulating hemocyte population and are responsible for the engulfment of small particles and dead cells, and they are able to secrete AMPs. Crystal cells contain copper-containing prophenoloxidase compounds, which oxidize phenolic molecules to produce melanin around invading pathogens and wounds. In uninfected larvae, the lamellocytes are present in small numbers in the late third instar stage, while otherwise healthy larvae do not contain them. Lamellocytes are only produced upon invasion of parasitoid wasps and form a multilayer capsule around the invading parasitic egg, with the help of plasmatocytes and crystal cells. Eventually, the capsule is melanized, and elevated levels of ROS (Reactive Oxygen Species) terminate the intruder [23].

## 3. How Are Bacteria Sensed by the Nervous System? 

Since detecting danger is key to the survival and success of all species, recent work has shown that animal nervous and immune systems cooperate to optimize danger detection. In animals, the accurate identification of chemicals followed by an internal assessment of sensory stimuli serves both to identify beneficial compounds and to avoid toxic ones [24]. In metazoans, chemoreception evolved two anatomically and functionally distinguishable systems, olfaction and taste, both of which have been implicated in the detection of bacteria. Several studies demonstrate that fly species’ detection of microbe-derived odors modifies their behavior either towards attraction or repulsion. As expected, these effects are highly species-dependent and are different—and sometimes opposite—at various stages of an animal’s life [25]. While adults and larvae are attracted to the volatile compounds produced by *Saccharomyces cerevisiae* and *Lactobacillus plantarum*, they are repelled by *Acetobacter malorum* in behavioral assays [26]. The attraction to yeast is governed by olfactory sensory neurons expressing the odorant co-receptor, called Orco, whereas the repulsion caused by bacteria is independent of it. When mixed with food, the opportunistic pathogen *Erwinia carotovora caratovora* (*E.c.c*) induces a blockage of larval food intake, which involves the Orco protein and the nociceptor TrpA1 [27]. In some cases, the bacterial compounds that interact with the sensory systems have been identified. *Drosophila* larvae can detect propionic and butyric acids, two short-chain fatty acids that are produced by many bacteria species, including the microbiota-associated *Lactobacilli* [28]. Propionic acid detection by odorant receptor 30a (Or30a) and Or94b+ neurons increase larvae feeding behavior, an appetite-stimulating effect not observed in adults or in the related species *S. suzukii*. Geosmin, a volatile odorant produced by some fungi and bacteria, acts as a potent repellent that can negate the fly’s innate attraction to vinegar. It is detected by a single class of neurons expressing the odorant receptor 56a [29]. Chemosensation is also used by flies to identify favorable feeding and egg-laying sites. *D. melanogaster* displays a strong oviposition aversion toward feces from carnivorous mammals, which contains a high rate of pathogenic bacterial taxa, but not toward herbivore dung, which is devoid of it. Or46a’s neuron-dependent detection of phenol produced by harmful bacteria explains this repulsion [30]. Inter-microbe metabolic exchanges that generate unique and quantitatively different volatiles add some complexity to bacteria detection by the fly sensory system. Hence, *Drosophila* prefers a *Saccharomyces–Acetobacter* co-culture to the same microorganisms grown individually and then mixed. Indeed, the bacteria–yeast interaction produces acetic acid whose detection by Or42b neurons is attractive to them [31]. The gut microbiota itself can also modify the microbial preferences of flies. While conventionally raised or axenic flies exhibit preference for Lactobacillus, fly preference for *Acetobacter* is primed by early-life exposure, and can override innate preference [32]. Although most studies implicate the olfactory system, the taste apparatus also plays a role in bacteria sensing and detection. Detection of bacterial lipopolysaccharide (LPS) by the esophageal bitter neurons via the TrpA1 (Transient receptor potential cation channel subfamily A member 1) receptor triggers feeding and oviposition avoidance [33]. LPS can also trigger grooming when applied to the wing margins of the flies, although the sensory system remains unknown in this case [34]. As one might expect, the behavior that flies adopt toward a given bacteria species requires the integration of multiple sensory modalities. When given the choice between a sugar solution and an *E.c.c* solution, flies are initially attracted to the bacteria and then, a few hours later, repelled by them. While the initial phase of attraction depends on Gr63a (Gustatory receptor) olfactory neurons, the repulsive phase is controlled by Gr66a bitter neurons. Moreover, bacteria being a food source for flies, their consumption causes the potentiation of bitter neurons allowing the establishment of the avoidance behavior [35]. These data highlight the key roles played by fly sensory neurons in detecting environmental bacteria and in initiating behaviors to either avoid bacteria-spoiled sources or, in contrast, to move towards them (Figure 2).

## 4. The Immune PGRP/NF-κB Module Plays an Important Role in Bacteria–Neuron Interactions 

Whereas some members of the Toll family and downstream transducers are implicated in the development of the fly central nervous system (CNS) [36,37,38], there is growing evidence that the second NF-kB pathway—the PGRP/IMD axis—mediates some of the interactions between bacteria and the cells that compose the fly nervous system. Indeed, by directly acting on a couple of brain octopaminergic neurons, bacteria-derived peptidoglycan causes a reduction in the number of eggs laid by infected females [39,40]. This effect requires functional PGRP and NF-κB signaling components, demonstrating that a unique bacterial cell wall constituent and a common host signaling cascade are used in immune cells to mount an immune response and, in brain neurons, to control fly behavior following infection. Some IMD pathway components are unexpectedly necessary to regulate presynaptic homeostatic potentiation (PHP) at the fly neuromuscular junction. This mechanism, which operates to stabilize synaptic activity in the nervous system, requires the PGRP-LC membrane receptor, and some downstream pathway components [41]. However, since PHP has no roles in antibacterial immunity, it is possible that PGRP-LC is activated at the synapse by an endogenous ligand [42]. PGRP proteins have also been implicated in regulating the balance between attraction and repulsion to bacteria. When given the choice, flies innately prefer the odor of pathogenic versions of two bacteria species, *E.c.c.* and *Pseudomonas entomophila,* over harmless mutant versions. However, this initial attraction turns into a lasting feeding suppression after ingestion, a behavioral adaptation that relies on the mushroom body and on the PGRP-LC and -LE functions in octopaminergic neurons [43] (Figure 1). In this case, the ligand that activates the PGRP proteins remains elusive, since there is no reason to believe that harmless and pathogenic bacteria differ in their PGN composition or structure. A recent report proposes an alternative model which states that this subsequent avoidance in bacteria comes from a reduction of the initial olfaction-dependent attraction. Infection-induced, unpaired cytokine expression in the intestine activates the JAK–STAT pathway in ensheathing glia. This signaling events trigger a glial cell metabolic reprogramming that, in turn, modulates olfactory discrimination and, hence, promotes the avoidance of bacteria-contaminated food [44] (Figure 2).

## 5. The NF-κB Pathway and Its Transcriptional Targets, the AMP, Regulate Other Behaviors in Flies

Several studies report unanticipated examples of how canonical immune genes may influence some of the neuronally controlled behavior in flies, although in most cases precise mechanistic insights are still missing. Along with other immune regulators, the transcription of NF-κB/Relish increases during sleep deprivation [45]. Consistently, NF-κB/Relish mutant flies show a reduced sleep period and, unlike their wild-type siblings, are unable to increase their sleep phase upon bacterial infection [46] (Figure 1 and Figure 2). The fact that both phenotypes can be corrected by an exogenous supply of NF-κB/Relish to adipocytes suggests that the Relish protein acts in a non-tissue, autonomous manner on the cells that control sleep. As mentioned above, the canonical NF-κB antibacterial pathway functions in octopaminergic neurons to regulate oviposition during bacterial infection. While our preliminary data suggest that canonical antimicrobial peptides (Diptericin, Cecropin…) do not mediate this effect (AM, LK, and JR, unpublished), other molecules with antimicrobial activity appear to be active in neurons. The level of Nemuri—a peptide with antimicrobial properties expressed in some brain neurons—is increased by sleep deprivation [47]. Its overexpression in neurons protects flies from infection by *Serratia marcescens* or *Salmonella pneumoniae*. Other experiments demonstrate that the expression of drosocin in neurons, or of metchnikowin in glial cells, enhances resilience to sleep deprivation [48]. Finally, fly mutant for Achilles, a rhythmically expressed neuronal gene, displays elevated levels of immune effectors, including AMPs [49]. As a result, flies are more resistant when exposed to bacteria. Memory is another neuron-dependent process in which immune genes have recently been implicated. Through the course of exploring how animals form long-lasting memories, it was found that the AMP Diptericin B (DptB) was induced in the head of flies following behavioral training that to produce long-term memory [50]. Targeted gene inactivation revealed that DptB activity was required to modulate long-term memory, although not in the neurons themselves but in the fat body surrounding the head (Figure 1). Further work will be required to identify the exact role played by AMPs, which are known to be very pleiotropic [51]. If, in some cases, AMPs play positive roles in the CNS, phenotype analysis of brains in which AMPs are overexpressed or brains of mutants with IMD-negative regulators (in which AMPs are expressed at abnormally high levels) indicate that they have detrimental effects for the host. Hyperactivation of innate immunity in the brain due to genetic mutations or bacterial injection causes neurodegeneration related to the neurotoxic effects of AMPs [52]. Ageing flies that show constitutive NF-κB-dependent AMP expression in glial cells suffer from progressive neurodegeneration and locomotion defects [53,54]. The increasing number of immune proteins and pathways involved in neuronal functions, and the clear benefits related to behavioral modification following exposure to microbes, cause more questions surrounding the precise delineation of the range of phenomena to be considered strictly as immune.

## 6. Octopamine, a Neuromodulator Implicated in Many Bacteria *Drosophila* Nervous System Interactions

In a few rare cases, the nature of the neuropeptides and the neural circuits that mediate bacteria–neuron interactions have been identified. One of the fly neuromodulators, the bioamine octopamine (OA), which controls diverse behaviors, such as learning, memory, or aggression, plays a fundamental role in such interactions [55]. In addition to the above-mentioned role in regulating egg-laying behavior of infected females, OA mediates some of the interactions between *Drosophila* and its endosymbiont *Wolbachia*. By comparing strains carrying or not carrying the symbiont, it was shown that *Wolbachia* can modify male aggressive behavior by modulating biosynthetic pathways and reducing neural OA levels [56]. A more recent study extends these data to the role of microbiota on fly aggression. Compared to their siblings carrying microbiota, germ-free males showed a substantial decrease in inter-male aggression, a phenotype that is rescuable by microbial recolonization [56] (Figure 2). These effects, which require the presence of the microbiota during a critical period of development, depend on OA biosynthesis, which is stimulated by the presence of bacteria in the fly gut. Other behaviors such as locomotion or courtship are not impacted by the presence of bacteria in the fly digestive tract. These results are at odds with a previous report showing that, when reared under axenic conditions, flies show greater locomotor activity than their siblings with intact microbiota [57]. In this case, it is proposed that xylose isomerase produced by bacteria such as *Lactobacillus brevis* is essential for maintaining normal locomotion in flies. Through mechanisms that still need to be understood, xylose isomerase exerts its effects by inactivating CNS octopaminergic neurons (Figure 2). The effect of the gut microbiota on fly behavior remains debated; however, another study shows that defensive behavior, sleep, locomotion, and courtship are weakly influenced by the absence of the microbiome [58]. On the contrary, flies in which the gut microbiota has been indirectly modified by the inactivation of *KDM5* methylase show abnormal social behavior [59].

## 7. When Blood Cells and Neurons Communicate to Respond to External Cues

The synergy between the immune and the nervous systems in the perception and the response of microbes is also found in the cellular part of immunity [60]. The number and properties of cells with phagocytic capacities vary in response to developmental and environmental cues, some of which are neuronal in origin [61,62]. The *Drosophila* sensory neurons contact hemocytes in hematopoietic pockets and regulate their proliferation, survival, and localization [63]. The TGF-β (Transforming Growth Factor) family ligand activin-β, which is expressed by peripheral sensory neurons, regulates hemocyte proliferation and adhesion. Activation or transient silencing of these neurons affects the number and location of resident hemocytes. On the other hand, signals coming from the environment can impact *Drosophila* hematopoiesis via neuronal activation. Activation of some fly olfactory neurons can trigger GABA (Gamma-aminobutyric acid) secretion by neurosecretory cells [64]. The activation of GABA metabotropic receptors expressed on hematopoietic progenitors regulates the balance between their maintenance and differentiation. While it is clear that the olfactory receptor Or42 is required for this process, the ligand(s) it senses remains unknown. A link has also been uncovered between CO_2_-sensing neurons and hematopoietic cells. Inactivation of these neurons leads to a hypoxia-inducible factor-α-dependent Unpaired3 production by downstream secondary order neurons [65]. In turn, these neurons release in the circulating blood the JAK/STAT pathway ligand, Unpaired3. By promoting insulin-like peptide-6 production by adipocytes, this hormone promotes the differentiation of crystal cells in the lymph gland. Since metabolically active microbes release various gases in their immediate environment, one might wonder whether bacterial infection directly activates these olfactory and gas-sensitive neurons that function upstream of hematopoietic differentiation.

## 8. Behavioral Immunity toward Parasitoid Wasps

Parasitoid wasps are vicious predators of *Drosophila* that, after puncturing the larvae with their sharp ovipositors, lay one egg inside them [66]. The developing wasps then feed on the larvae’s tissues from the inside, then finally hatch from the pupal cases, instead of the flies. In nature, the rate of parasitism is estimated to be approximately 90%, resulting in great selective pressure on *Drosophila* populations [66]. However, *Drosophila* larvae are not defenseless. While some defense mechanisms only implicate immune mechanism such as the melanotic encapsulation of the wasp eggs by lamellocytes [67,68], others rely on the fly CNS. To escape the wasps’ attacks, larvae perform a series of stereotyped movements, depending on the point of attack and whether or not the cuticle is penetrated. This nociceptive response is mediated by class IV neurons that are required for mechanical nociception [69] (Figure 3). Alongside the cellular immune response and larvae rolling behavior, recent studies have uncovered several other behavioral strategies that the fly species has developed to protect its offspring from parasitoid wasps. The first defense is the avoidance behavior triggered in both adults and fly larvae by wasps’ odors. This innate avoidance response is mediated by specific olfactory receptor neurons that co-express two odorant receptors (Or49a and Or85f) tuned to detect wasp semiochemicals [70]. In addition to what could be considered a long-range aversion to wasps, other behavioral responses that rely on visual and olfactory cues and depend on the environmental context can be triggered [71]. First, the sight and smell of wasps causes a rapid decline in oviposition through egg retention, coupled with apoptosis via effector caspases at a specific stage of oogenesis [70,72]. These parasitoid-induced changes in germ cell development are induced by suppression of neuropeptide F (NPF) signaling in the brain, which occurs following the sight of wasps [73,74]. The oviposition depression is triggered when the fly sees the wasp. Strikingly, this instructed animal (teacher) can in turn repress the oviposition of a naïve fly (student) if they can establish visual contact. Moreover, depression of NPF signaling has been linked to a greater tolerance and preference for ethanol [73]. This enhanced taste for ethanol guides a second behavioral immune response, consisting of a shift in the preference of female flies toward alcohol-enriched oviposition sites. Indeed, as *D. melanogaster* larvae have a higher tolerance to ethanol than their parasitoids, females choose to preemptively medicate their offspring by favoring oviposition sites with toxic levels of alcohol [73,75]. However, such an interpretation has been debated [76]. Strikingly, this predisposition for ethanol-rich food can be inherited by up to five generations, through an epigenetic reprogramming that requires an NPF-induced activation of caspases in the female germline [77]. Another transgenerational effect of wasp exposure has been reported by a later study. Specifically, the offspring of exposed females have a faster rate of immune cell production, which correlates with better survival to wasp infection [78]. Transcriptional analysis of oocytes revealed maternal mRNA’s substantial role in shaping the immune phenotype inherited by the offspring [78]. Thus, it could be hypothesized that, in response to the predatory threat, females halt costly egg production and allocate their resources to finding an uninfected oviposition site better suited to producing fewer offspring with enhanced immunity and which are better suited to withstanding wasp infection. Following this logic, one would expect that the presence of a parasitoid would delay mating. Counterintuitively, a recent study shows that the sight of the wasp causes an acceleration in mating behaviors [79]. A possible explanation for this unexpected phenotype is that, while seizing the opportunity to mate by speeding up mating, the female saves time and resources to then search for an egg-laying site away from the threat. The sight of wasps induces an upregulation in the fly nervous system of a 41-amino acid micropeptide whose function is essential to the behavioral response of the fly [79]. Finally, an increasing number of studies have explored the long-term effects of parasitoid exposure. Indeed, the oviposition decline and the preference for ethanol-enriched food are maintained for a few days after exposure to the wasp [73,80]. For these behavioral changes to persist, several long-term memory genes are required, as well as synaptic transmission in the mushroom bodies, which are the brain center for memory and learning [71]. Surprisingly, predatory threat also triggers the social transfer of information between experienced and naïve individuals. Exposed flies become teachers and transmit through visual cues the information to naive flies, which in turn reduces oviposition through the same apoptotic mechanism [71]. While the memory of a threat is retained in both exposed and naïve-educated flies, only the former has the ability to teach. Thus, by limiting social the learning of oviposition reduction to an exposed fly and its neighbors, the fitness costs of widespread adoption of this behavior are averted (Figure 3).

## 9. Concluding Remarks

Having gathered some of the recent studies that elucidate with molecular details the links between immunity, protection of the host, and detection of microbes in *Drosophila*, it appears that these complex systems are highly intricate, and not only cooperate, but also share strategies and functions. The coevolution of these two functions for the sake of the host can be seen as a highly efficient cooperation from the beginning, with specialties appearing along the way. Although the speed of progress in this emerging field is astonishing, we are still far away from understanding how flies, as in every other organism, integrate such a large variety of cues coming from diverse surrounding organisms to adopt the appropriate behaviors.

## Figures and Tables

**Figure 1 cells-10-02350-f001:**
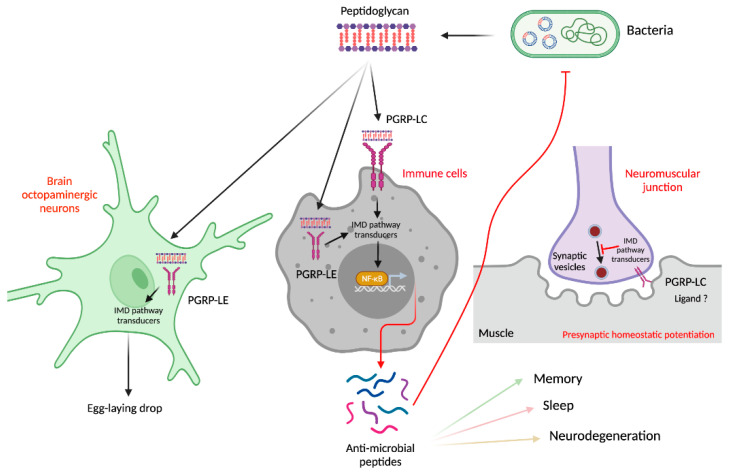
Dual role of the PGN/PGRP/NF-κB pathway in immune and neuronal cells. (See text for details).

**Figure 2 cells-10-02350-f002:**
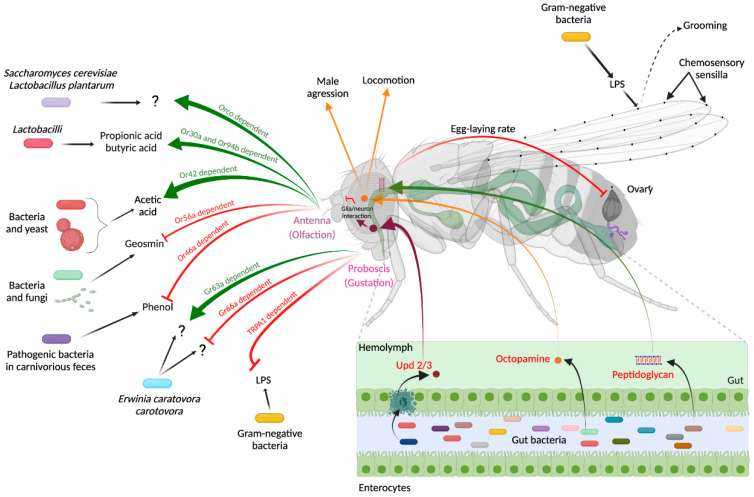
Mechanisms of interactions between microbes and the *Drosophila* nervous system. (See text for details).

**Figure 3 cells-10-02350-f003:**
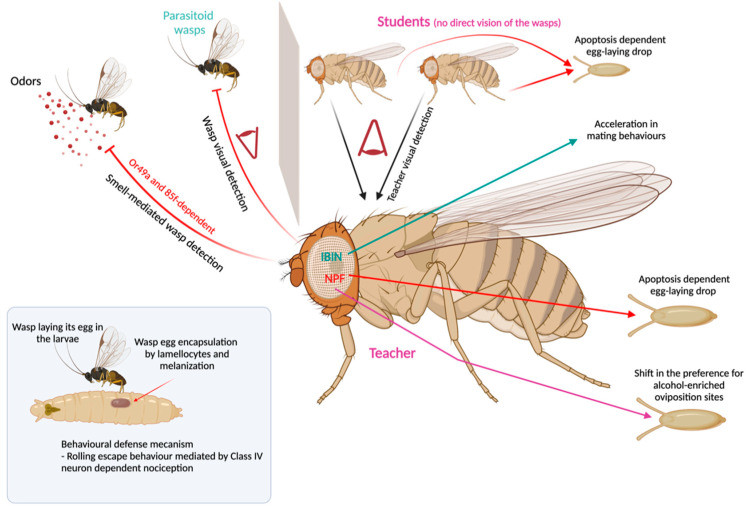
*Drosophila*–parasitoid wasp interactions (see text for details).

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
