# Peer review of "Impact of Microorganisms and Parasites on Neuronally Controlled Drosophila Behaviours"

_cells, 2021, doi:10.3390/cells10092350_

Round 1
Reviewer 1 Report
This is a well-written, timely assessment of the importance of the neuronal-immune axis and its importance during host-pathogen interaction. I have only two questions/comments regarding the content and a few suggestions regarding style/grammar.
Content:
Line 77: there is also evidence for lamellocyte production after wounding
(see: Sterile wounding is a minimal and sufficient trigger for a cellular immune response in Drosophila melanogaster.Márkus R, Kurucz E, Rus F, Andó I. Immunol Lett. 2005 Oct 15;101(1):108-11. doi: 10.1016/j.imlet.2005.03.021)
The ethanol preference described in lines 283 ff seems to have been a matter of debate (see: Ethanol confers differential protection against generalist and specialist parasitoids of Drosophila melanogaster. Lynch ZR, Schlenke TA, Morran LT, de Roode JC.PLoS One. 2017 Jul 12;12(7):e0180182. doi: 10.1371/journal.pone.0180182. eCollection 2017). I wonder what the latest state is? Should that be mentioned?
Style:
Line 25; my suggestion: life-threatening
Line 32; my suggestion: microbe-derived (also in other places)
Line 43; my suggestion: regulating both the neuronal and the immune system
Line 48; my suggestion: are still missing
Line 98; my suggestion: bacterial compounds (also in other places)
Line 103: perhaps define orexigenic (appetite-stimulating?)
Line 109; my suggestion: bacterial taxa
Line 116; my suggestion: conventionally raised
Line 126: ; my suggestion: While the initial phase…
Line 138: ; my suggestion: there is growing evidence…
Line 143; my suggestion: bacterial cell wall…
Line 150; my suggestion: no role in bacterial immunity
Line 183; my suggestion: enhances resilience…
Line 224; my suggestion: that need to be understood
Line 237/238: 2 prepositions: by in
Line 261; my suggestion: While some defense mechanisms
Line 306: define MB
Line 310; my suggestion: which in turn reduces…
Line 324; my suggestion: far away from understanding how the fly….
Author Response
Response to reviewer:
Line 77: there is also evidence for lamellocyte production after wounding
(see: Sterile wounding is a minimal and sufficient trigger for a cellular immune response in Drosophila melanogaster.Márkus R, Kurucz E, Rus F, Andó I. Immunol Lett. 2005 Oct 15;101(1):108-11. doi: 10.1016/j.imlet.2005.03.021)
We agree but our review deals with the host response to bacteria. This is why we did not include it.
The ethanol preference described in lines 283 ff seems to have been a matter of debate (see: Ethanol confers differential protection against generalist and specialist parasitoids of Drosophila melanogaster. Lynch ZR, Schlenke TA, Morran LT, de Roode JC.PLoS One. 2017 Jul 12;12(7):e0180182. doi: 10.1371/journal.pone.0180182. eCollection 2017). I wonder what the latest state is? Should that be mentioned?
We have mentioned it.
Style:
Line 25; my suggestion: life-threatening
Line 32; my suggestion: microbe-derived (also in other places)
Line 43; my suggestion: regulating both the neuronal and the immune system
Line 48; my suggestion: are still missing
Line 98; my suggestion: bacterial compounds (also in other places)
Line 103: perhaps define orexigenic (appetite-stimulating?)
Line 109; my suggestion: bacterial taxa
Line 116; my suggestion: conventionally raised
Line 126: ; my suggestion: While the initial phase…
Line 138: ; my suggestion: there is growing evidence…
Line 143; my suggestion: bacterial cell wall…
Line 150; my suggestion: no role in bacterial immunity
Line 183; my suggestion: enhances resilience…
Line 224; my suggestion: that need to be understood
Line 237/238: 2 prepositions: by in
Line 261; my suggestion: While some defense mechanisms
Line 306: define MB
Line 310; my suggestion: which in turn reduces…
Line 324; my suggestion: far away from understanding how the fly….
We have included these modifications.
Reviewer 2 Report
In this review, Montanari and Royet cover an interesting and emerging field of neuro-immune interaction and comprehensively summarize the latest progress in the field. Due to the complexity, it is challenging to understand intricate and direct communications between the two systems and Drosophila with its powerful genetic and genomic tools has provided an excellent platform to uncover new mechanisms and paradigms in the neuro-immune link. Recent findings in the Drosophila immune and sensory systems, respectively, are well-reviewed, and novel components in the neuro-immune control, including PGRP, AMP, and OA, are carefully incorporated. Overall, this review will not only provide in-depth insights into the links between neurons and the innate immune system but together throw numerous questions to researchers in the related field which help to expand the field. It’s a well-written review and I have only a few comments to reconsider before publication. 1. Fig 3 is not as intuitive as the other two figures. It could be better structured to illustrate behavioral changes of flies upon the fly-wasp interaction as described in section 8. 2. There are typos to correct. Line 91: sometime - sometimes Line 138: they – there Line 162: this – these Line 163: in tuns – in turn Line 233: an additional period before the reference Line 170: NfkB Relish – NF-kB/Relish Figure 1: lamelocytes - lamellocytes 3. Overall American English and British English are mixed. This could be confusing and needs clarification. For example, Odour/odor; behaviour/behavior; behavioural/behavioral; haemocytes/hemocytes etc.Author Response
In this review, Montanari and Royet cover an interesting and emerging field of neuro-immune interaction and comprehensively summarize the latest progress in the field. Due to the complexity, it is challenging to understand intricate and direct communications between the two systems and Drosophila with its powerful genetic and genomic tools has provided an excellent platform to uncover new mechanisms and paradigms in the neuro-immune link. Recent findings in the Drosophila immune and sensory systems, respectively, are well-reviewed, and novel components in the neuro-immune control, including PGRP, AMP, and OA, are carefully incorporated. Overall, this review will not only provide in-depth insights into the links between neurons and the innate immune system but together throw numerous questions to researchers in the related field which help to expand the field. It’s a well-written review and I have only a few comments to reconsider before publication. 1. Fig 3 is not as intuitive as the other two figures. It could be better structured to illustrate behavioral changes of flies upon the fly-wasp interaction as described in section 8. 2. There are typos to correct. Line 91: sometime - sometimes Line 138: they – there Line 162: this – these Line 163: in tuns – in turn Line 233: an additional period before the reference Line 170: NfkB Relish – NF-kB/Relish Figure 1: lamelocytes - lamellocytes 3. Overall American English and British English are mixed. This could be confusing and needs clarification. For example, Odour/odor; behaviour/behavior; behavioural/behavioral; haemocytes/hemocytes etc.
We have made the modification requested. As for the figure related to wasp infection, it is difficult to illustrate the behaviour changes upon wasp detection. We could not come up with a better idea.